# Strengthening Cracked Steel Plates with Shape Memory Alloy Patches: Numerical and Experimental Investigations

**DOI:** 10.3390/ma16237259

**Published:** 2023-11-21

**Authors:** Zhiqiang Wang, Libin Wang, Qiudong Wang, Bohai Ji, Jie Liu, Yue Yao

**Affiliations:** 1College of Civil Engineering, Nanjing Forestry University, Nanjing 210037, China; wzq010302@163.com (Z.W.); jhwlb@163.com (L.W.); jieliu@outlook.com (J.L.); 2Faculty of Architecture and Civil Engineering, Huaiyin Institute of Technology, Huai’an 223001, China; 3College of Civil and Transportation Engineering, Hohai University, Nanjing 210098, China; bhji@hhu.edu.cn; 4College of Construction Engineering, Jiangsu Open University, Nanjing 210000, China; jsouyaoyue@163.com

**Keywords:** steel plate, fatigue crack, strengthening, SMA, fatigue life

## Abstract

To investigate the retarding effect of bonding the shape memory alloy (SMA) patches on crack propagation in steel plates, both numerical and experimental analyses were conducted in the present study. A compact tension (CT) model was developed to clarify the feasibility of bonding the SMA patch to the reinforcement of the mode Ⅰ, mode Ⅱ, and mode Ⅲ cracks. On this basis, parametric analysis was conducted to investigate the strengthening parameters, i.e., the bonding area, the thickness, and the strengthening angle of the SMA patch. Subsequently, fatigue tests on the unreinforced steel plate and cracked steel plate strengthened by the SMA patches were conducted. The monitored stress variation, crack propagation behavior, and fatigue fracture surfaces were analyzed. Findings are meaningful to the application of the SMA reinforcement method in practical engineering.

## 1. Introduction

Steel structures (e.g., marine steel structures and steel bridges) are easily subjected to fatigue cracking under cyclic dynamic loading [1]. Fatigue cracks are permanent damage to steel members, which significantly threaten the durability, service performance, and safety of steel structures [2]. It is necessary to take effective measures to strengthen or repair the cracked steel components once the crack has been detected.

Many strengthening measures have been developed to retard the crack propagation in steel structures, such as drilling stop-hole [3,4], hammer peening [5,6], attaching additional steel patches [7], etc. In bridge engineering, bonded/bolted steel plate reinforcement and bonded carbon/glass fiber reinforced polymer (CFRP/GFRP) reinforcement have been widely used [8,9,10,11]. In terms of the strengthening methods by attaching additional elements, the theoretical basis of these solutions is that a second load path could be obtained by the additional elements attached to the cracked steel components. Subsequently, the load transferred by the cracked steel component could be considerably reduced, thus reducing the stress intensity factor (SIF) at the crack tip and retarding the crack propagation. On this basis, it is obvious that the crack propagation rate would be further reduced if the prestress was applied to the attached element. Furthermore, it has already been proved that a complete crack arrest could be achieved in the presence of the introduction of an adequately high level of prestressing.

However, cracked steel components are usually relatively small, and it is rather difficult to conduct the prestressing process of attached elements (e.g., fiber-reinforced polymer (FRP) plates). In recent years, the investigation and application of shape memory alloys (SMA) in civil engineering have raised great concern [12,13]. In contrast to the traditional strengthening method (e.g., attaching the steel or FRP elements), the prestressing effect could be achieved by taking advantage of the SMA materials. The SMA material has a unique thermomechanical response in which deformed SMA patches can return to their original geometry after thermal activation [14]. Once the deformed SMA element (before thermal activation) is attached to the cracked component, the prestress, induced by the recovery forces after thermal activation, could be applied to the cracked component; thus, the crack could be retarded or even arrested by the prestressing effect of the SMA element [15].

Some investigations have been conducted to investigate the performance of cracked components reinforced by the SMA materials. Wang et al. [16] conducted fatigue tests on cracked steel plates strengthened by CFRP and iron-based SMA (Fe-SMA) strips, revealing that the bonded prestressed Fe-SMA strips are much more effective than CFRP strips, extending the fatigue crack growth life by a factor of 3.51. Izadi et al. [17] revealed that a prestressing level within the range of 330–410 MPa could be achieved, resulting in compressive stresses within the range of 35–72 MPa in the cracked steel plates. Results also demonstrated that considerable compressive stresses could be applied to the cracked steel plate, reducing the tensile stresses and SIFs at the crack tip, subsequently resulting in a significant decrease in the crack propagation and a complete crack arrest in some cases. Chen et al. [18] investigated the durability of steel structures strengthened by Fe-SMAs when subjected to harsh service conditions via fatigue tests. Wang et al. [19] investigated the behavior of the Fe-SMA-to-steel bonded joints through lap-shear tests, revealing that no debonding or degradation was observed for the adhesively bonded Fe-SMA-strengthened specimen and demonstrating the reliable performance of the strengthening solution under service loads. Vujtech et al. [20] conducted an application of an iron-based SMA (Fe-SMA) for the prestressed strengthening of a bridge, revealing that the recovery stress of the Fe-SMA strips resulted in a compressive stress of approximately −33 MPa in the lower flange of the bridge girder. Additionally, the long-time monitoring showed that the main loss of the prestressing force caused by relaxation occurred within the first 30 days after activation and was approximately 20% of the original prestress. Li et al. [21] conducted a systematic study on the static behavior of adhesively-bonded Fe-SMA-to-steel joints in applications adopting iron-based Shape Memory Alloys (SMAs).

At the same time, CFRP/SMA composite patches have been proposed to improve the reinforcement effect on cracked components further. For instance, Qiang et al. [22] proposed to repair the fatigue crack at the diaphragm arc-shape cutout by employing SMA/CFRP composite patches, indicating that the fatigue notch factor was reduced by 30.76% after bonding SMA/CFRP composite patches, and the initiation and propagation of fatigue cracks could be effectively postponed. Fatigue tests conducted by Deng et al. [23] indicated that the crack propagation rate could be considerably reduced by applying the SMA/CFRP composite strengthening, and the load capacity and stiffness of notched steel beams could be significantly improved. Kean et al. [24] performed a numerical study of the fatigue life of SMA/CFRP patches retrofitted to central-cracked steel plates, indicating that the increase in the Young’s modulus, prestress level, and width could significantly improve the fatigue life of central-notched steel plates. Similar investigations have also been reported [25,26].

In general, some investigations have been conducted, focusing on the fatigue performance of cracked steel components strengthened by the SMA patches or the SMA/FRP composited layers. The retarding effect of attaching SMA elements on cracked steel components is proved. However, the mode Ⅰ crack condition is considered in most of the investigations, while fewer investigations have focused on the retarding effect on different cracking types, which is the focus of the present study. In terms of the experimental investigations, the crack propagation behavior and the fatigue life before and after strengthening are the main focus. However, the stress variation in the SMA and reinforced steel component and the fatigue fracture surfaces of the reinforced steel plates were not clear.

In this study, numerical and experimental investigations were performed to illustrate the performance of cracked steel plates strengthened by Fe-SMA patches. CT models were first developed to clarify the retarding effect on different types of fatigue cracks. On this basis, parametric analyses were conducted to investigate the effects of strengthening area, thickness, and position of the Fe-SMA patch and the strengthening angle. Finally, fatigue tests were conducted to investigate the stress variation, crack propagation, and fatigue failure characteristics of cracked steel plates strengthened by the Fe-SMA patches.

## 2. CT Model-Based Feasibility Analysis

### 2.1. Finite Element Model

The fatigue crack could be divided into three basic types: mode Ⅰ cracks subjected to in-plane tension, mode Ⅱ cracks subjected to in-plane shear, and mode Ⅲ cracks subjected to out-plane shear. Under different loading conditions, the SIF at the crack tip could be adopted to identify the mode Ⅰ, mode Ⅱ, and mode Ⅲ cracks, namely labeled *K*_Ⅰ_, *K*_Ⅱ_, and *K*_Ⅲ_. To investigate the feasibility of attaching Fe-SMA reinforcement to different cracking types, the CT model recommended by the American Society for Testing and Materials (ASTM) was developed by ABAQUS 6.14 Documentation (Dassault Systemes Simulia Corp, Providence, RI, USA) since the basic three types of fatigue cracks could be easily simulated. The extended finite element method (XFEM) embedded in ABAQUS was employed to obtain the SIF at the crack tip. A mesh size of 5 mm was employed to model the CT model, and the Fe-SMA patch was modeled with a mesh size of 2 mm [27]. The element type of C3D8R, including 8 Gaussian integral points with a size of 0.5 mm, was used to deal with the stress concentration at the crack tip, considering the accuracy and efficiency of the calculation. The geometric sizes of the CT model are plotted in Figure 1. The thickness of the CT model was set to 5 mm. The crack was simplified to be a rectangle with a length of 50 mm.

Steel and Fe-SMA are usually strain-hardening materials; thus, elastic/plastic properties should be considered in practical engineering. However, in this section, the CT model was considered to be in a linear elastic state, and the steel and Fe-SMA were simply assigned a linear elastic constitutive relation. The elastic modulus and Poisson’s ratio of the steel were set to be 2.06 × 10^5^ MPa and 0.3, and values of the Fe-SMA patch were set to be 1.70 × 10^5^ MPa and 0.36, respectively. It is difficult to simulate the thermal activation process of the Fe-SMA patch using finite element (FE) modeling. As an alternative method, the prestressing level of the Fe-SMA patch was modeled by introducing a predefined stress field in the initial step. The predefined stress field was then redistributed in the subsequent load step in ABAQUS, leading to the compressive stress in the strengthened component and a prestressing loss in the Fe-SMA patch owing to the contraction of the strengthened component. For more details regarding the modeling of the prestressing level of the SMA patch, please refer to [27]. In the present study, a prestress level of 0.075 MPa was employed for qualitative analysis.

### 2.2. Simulation of Different Fatigue Types

To simulate the mode Ⅰ crack, the uniform area loads with a magnitude of 10 MPa were applied at the upper and lower semicircular surfaces of the loading hole (see Figure 2a). The same loads were applied at the left and right semicircular surfaces to simulate the mode Ⅱ crack. At the same time, in terms of the CT model simulating the modes Ⅰ and Ⅱ cracks, the displacement of nodes at the top and bottom surfaces was fixed at the thickness direction to avoid the bulking effect. To simulate the mode Ⅲ crack, the loading hole was not simulated, and a uniform area load of 0.4 MPa was applied at the corresponding surface (see Figure 2a). Additionally, nodes at the right end of the model were fixed. Based on the J integral and XFEM, the SIF corresponding to the mode Ⅰ, mode Ⅱ, and mode Ⅲ cracks could be calculated. Since the crack was simplified to be a rectangle penetrating along the thickness direction of the CT model, the SIF at the middle point of the crack front was extracted, as plotted in Figure 2b. For the mode Ⅰ crack, the value of SIF *K*_Ⅰ_ was calculated to be 390 MPa·mm^1/2^, while the values of SIF *K*_Ⅱ_ and *K*_Ⅲ_ could be neglected. For the mode Ⅱ crack, the value of SIF *K*_Ⅱ_ was calculated to be 124 MPa·mm^1/2^, while the values of SIF *K*_Ⅰ_ and *K*_Ⅲ_ could be neglected. Similarly, the value of SIF *K*_Ⅲ_ was calculated to be 108 MPa·mm^1/2^ for the mode Ⅲ crack, while the values of SIF *K*_Ⅰ_ and *K*_Ⅱ_ could be neglected. The results indicate that the mode Ⅰ, mode Ⅱ, and mode Ⅲ cracks could be well simulated by the developed CT models.

### 2.3. Variation in the SIFs

To investigate the retarding effect of attaching the Fe-SMA patch on the mode I, mode II, and mode III cracks, the CT model strengthened by the Fe-SMA patch was simulated. For the feasibility analysis, the geometric size of the Fe-SMA patch was set to be 30 × 55 × 0.5 mm, as shown in Figure 3a. The influence of the geometry of the Fe-SMA patch was to be discussed in the following Section 3. The SIFs corresponding to the CT model and the CT model strengthened by the Fe-SMA patch were compared, as plotted in Figure 3b. The SIF corresponding to the CT model strengthened by the steel patch was also added for comparison. The geometry of the steel patch was the same as the Fe-SMA patch, and the elastic modulus and Poisson’s ratio were the same as the material properties of the CT model. It can be seen from Figure 3b that, after bonding the steel and Fe-SMA patch, the SIF decreased by 12.1% and 37.3%, respectively. It indicates that attaching the Fe-SMA patch has a better reinforcement effect than attaching the steel patch. Regarding the mode Ⅱ and mode Ⅲ crack, the SIF has only decreased by 9.2% and 10.2%, respectively. It indicates that other strengthening methods should be taken for retarding the propagation of mode Ⅱ and mode Ⅲ crack instead of attaching the Fe-SMA patch.

## 3. Parametric Analyses on the Strengthening Method

### 3.1. Effect of the Bonding Area

In practical engineering, it is difficult to bolt the small-sized Fe-SMA patch to the cracked component. Alternatively, the Fe-SMA patch could be bonded to the cracked component using a cementing compound, such as that employed in the present study. In this case, the strengthening effect might be affected by the bonding area (as illustrated in Figure 4a). To investigate the effect of the bonding area, half of the length of the bonding area along the lengthwise direction of the Fe-SMA patch (labeled *x*) was set to be 5.5 (1/10 of the length of the Fe-SMA patch), 11.0, 16.5, 22.0, and 27.5 mm (half of the length of the Fe-SMA patch), respectively. The width of the bonding area was the same as that of the Fe-SMA patch. The mode Ⅰ crack was simulated, and the results were plotted in Figure 4b. The value of the SIF *K*_Ⅰ_ decreased from 305.3 to 298.5 MPa·mm^1/2^ as the value of *x* increased from 5.5 to 22.0 mm. The smallest value of the SIF *K*_Ⅰ_ (257.6 MPa·mm^1/2^) was obtained when the Fe-SMA patch was completely bonded to the CT model (i.e., *x* = 27.5 mm). However, in terms of the experimental investigations, it is not suggested that the Fe-SMA patch be completely bonded to the cracked component. The reason is that the bonding property of the bonding layer within the thermally activated region might be affected by the high activation temperature; however, it is difficult to quantify such an effect. To eliminate the influence of uncertain factors on the experimental results, a limited area of the Fe-SMA patch was bonded to the cracked component in the present study, as to be introduced in the following Section 4.

### 3.2. Effect of the Thickness of the SMA Patch

The prestressing force could be generated by the thermomechanical response of the Fe-SMA patch. To investigate the effect of the thickness of the Fe-SMA patch on the retarding effect, the thickness was set to be 0.5, 1.0, and 1.5 mm, respectively. The mode Ⅰ crack was simulated, and the results were plotted in Figure 5. It could be seen that the value of the SIF *K*_Ⅰ_ decreased from 305.3 to 161.1 MPa·mm^1/2^ while the thickness of the Fe-SMA patch increased from 0.5 to 1.5 mm. It can be concluded that the thicker the Fe-SMA patch, the greater the prestressing force.

### 3.3. Effect of the Strengthening Angle

Due to the possible manual error during the bonding process, the lengthwise direction of the Fe-SMA patch may not be perpendicular to the crack propagation direction. To clarify such an effect, the strengthening angle (as illustrated in Figure 6a) was set to be 0°, 15°, 30°, and 45°, and the numerical results were plotted in Figure 6b. Similarly, the mode Ⅰ crack was also simulated here. It could be seen that the SIF *K*_Ⅰ_ increased from 305.3 to 351.5 MPa·mm^1/2^ while the strengthening angle (*θ*) varied from 0° to 45°. The results are readily comprehensible as a horizontal component of the prestressing force (parallel to the crack propagation direction) will be generated once the strengthening angle is greater than zero. Nevertheless, the SIF *K*_Ⅰ_ has increased by merely 1.7% (increased from 305.3 to 310.4 MPa·mm^1/2^) while the strengthening angle (*θ*) varies from 0° to 15°. Therefore, it is acceptable that the bonded Fe-SMA patch is not strictly perpendicular to the crack propagation direction in practical engineering.

## 4. Experimental Investigations

### 4.1. Specimens and Fe-SMA Patches

A series of fatigue tests on unreinforced and reinforced steel plates with artificial cracks was performed to investigate the efficiency of retarding the fatigue crack propagation by bonding the Fe-SMA patches. Three specimens were included in the fatigue tests: a bare steel plate was tested to obtain the reference data of the crack propagation behavior of the steel plate, labeled SP-1; two cracked specimens were strengthened by the prestrained Fe-SMA patches to obtain the data of crack propagation behavior after strengthening, labeled SP-2 and SP-3. The bare steel plates were manufactured by the China Railway Baoqiao (Yangzhou, China) Co., Ltd., and the Fe-SMA patches were manufactured by Suzhou Haichuan Rare Metal Products Co., Ltd. (Suzhou, China). The material properties of the steel and Fe-SMA were listed in Table 1 and Table 2, respectively. Notably, the material Q345qD possesses the same properties and composition as ASTM A572 [28] Gr 50 steel. The main geometric sizes of the steel plate and Fe-SMA patch are presented in Figure 7. The thickness of the steel plate and Fe-SMA patch were 14 and 3 mm, respectively.

To generate a fatigue crack at the objective position of the steel plate, each steel plate was cut first to induce a crack in the middle (see Figure 7a), and the crack length was 120 mm. Subsequently, the steel plate with an artificial crack was pre-tested under cyclic fatigue loading to generate real fatigue cracks at both ends of the artificial crack. During this stage, two strain gauges were arranged at both sides of the artificial crack, 5 mm away from the crack tip, to monitor the initial crack propagation. Once the strain gauge was broken, indicating that the fatigue crack had propagated below the strain gauge, the crack length was measured every five minutes. Once the fatigue crack at one side had propagated to 15 mm, the pre-test was stopped. Then, two cracked steel plates were strengthened by the deformed SMA patches at both sides of the fatigue cracks. In the present study, since the steel plate with artificial crack, fatigue load, and boundary conditions were symmetric, the propagation of fatigue cracks at two tips of the artificial crack was almost the same during the pre-testing process. When the fatigue crack at one side has propagated to 15 mm, the fatigue crack at the other side has commonly propagated to a length between 14 and 15 mm. In general, the total length of the fatigue crack after pre-testing was about 150 mm.

Before being adopted for strengthening, the undeformed Fe-SMA patches were annealed to obtain the best shape memory effect: the Fe-SMA patches were heated at 900 °C for 20 min, then removed and naturally cooled to room temperature. The prestressing force (i.e., the recovery force) is related to the prestrain of the Fe-SMA patch. Before bonding to the cracked steel plate, the Fe-SMA patches were prestrained to a strain of 5% and then unloaded to a free-stress state at room temperature. The clamped area of the Fe-SMA patches is shown in Figure 7b. The clamped area was fixed to the anchorage, and the rest part (i.e., 30 mm in the lengthwise direction) was stretched to 31.5 mm in this study.

### 4.2. Activation of the Fe-SMA Patch

Regarding the specimens labeled SP-2 and SP-2, the prestrained Fe-SMA patches were adhesively bonded to the cracked steel plate. Notably, only the clamped area (see Figure 7b) was bonded to the steel plate, while the deformed area remained unbonded. A two-part epoxy adhesive (typed Ergo 1309) was adopted to bond the Fe-SMA patch to the cracked steel plate. During the fatigue loading, no debonding failure of the adhesive layer was detected, indicating that this type of two-part epoxy adhesive could be adopted for experimental investigation.

After strengthening, a ceramic heating element (produced by Shanghai Yidu Electronics Co., Ltd., Shanghai, China) was adopted to activate the prestrained Fe-SMA patches, as shown in Figure 8. The maximum temperature heated by the ceramic heating element is 250 °C. The temperature along the Fe-SMA patch was monitored by a contact temperature detector (typed TM902C) (see Figure 8). As soon as the temperature in the Fe-SMA patch reached the available activation temperature of 250 °C, the electric current was cut off, and the thermal-activated Fe-SMA patch was naturally cooled down to room temperature.

Before thermal activation, a strain gauge (typed 120-3AA) was arranged at the surface of the Fe-SMA patch to monitor the variation in strain during the heating process, and the monitored strain data were plotted in Figure 9. It can be seen that considerable compressive strain was induced during thermal activation, and the maximum compressive strain was 690 με (corresponding to a compressive stress of 119.37 MPa) after heating for approximately 20 s. Notably, the real compressive strain induced by the shape memory effect was greater than 690 με because the measured results were significantly affected by the thermal expansion (generating tensile strain) of the strain gauge and the adhesive layer bonding the strain gauge to the Fe-SMA patch. It could be observed that, after heating for 20 s, the compressive strain gradually reduced and then turned into tensile. Subsequently, the strain gauge was debonded because of continuous heating. Generally, the monitoring results indicated that considerable compressive stress (greater than 119.37 MPa) was induced by the shape memory effect of the Fe-SMA patch; thus, considerable prestress was applied to the cracked steel plate for retarding the fatigue crack propagation.

### 4.3. Fatigue Test Setup

In the process of the fatigue pre-test, eight strain gauges (labeled from G1 to G8) were arranged at the extension of the artificial crack, as shown in Figure 10a. The resistance of the strain gauge is 120 Ω, and the sensitivity ratio is 2.0 ± 1.0%. The G4 and G5 were 5 mm away from the crack tip, and G6 (G3), G7 (G2) and G8 (G1) were 60, 70, and 80 mm away from the crack tip, respectively. Once G4 or G5 was broken, indicating the fatigue crack propagating below the strain gauge, the crack length was measured every five minutes. The pre-test was stopped while the fatigue crack at one side propagated to 15 mm. Then, the G4 and G5 were removed, and the corresponding position was polished. Subsequently, the SMA patches were bonded to the cracked steel plate, and the other two strain gauges were bonded to the activated SMA patches at the same position (i.e., 5 mm away from the artificial crack tip).

For applying the fatigue load, one side of the specimen was bolted to the rigid frame, and the other was bolted to the fatigue test machine (see Figure 10b). A bending-type load was then generated by the rotation of an eccentric block inside the vibrator. The G8 was selected as the reference point, and the applied stress range was controlled to be 180 MPa (with a stress ratio of −1). For SP-1, SP-2, and SP-3, the loading frequency was 31.5, 32.7, and 33.0 Hz, respectively. Once the fatigue crack had propagated to the lateral side of the specimen, the fatigue test was stopped. The fatigue crack propagation was recorded, and the fracture surface was also obtained, as to be introduced in the following section.

## 5. Test Results and Discussions

### 5.1. Strain Variation during Fatigue Testing

Given the large amount of monitored strain data and the similarity of the strain variation among the three specimens, the strain data of SP-2 were plotted for analysis. The strain gauges G4 and G5 were employed to monitor the strain at the surface of SMA patches, as plotted in Figure 11a,b. It could be seen that the surface strain was up to 700 με while the fatigue crack propagated below the SMA patch. Subsequently, a sharp decline in the strain-time curve was observed, and the subsequent strain was compressive. Integrating the experimental phenomenon, it was found that the occasion for the sharp decline corresponded to the fact that the fatigue crack propagated to the edge of the SMA plate. Therefore, it could be supposed that the fatigue load transferred by the SMA patch was significantly reduced once the fatigue crack propagated to the edge of the SMA patch. Meanwhile, since the prestress force (i.e., the recovery force) remained stable, the monitored strain turned into compressive. It could also be concluded that the strain variation characteristics bonded to the SMA patch could be employed to predict whether the fatigue crack has propagated to the edge of the SMA patch. Additionally, the strain-time curve of G4 and G5 stayed out to the end of the fatigue test, indicating that there was no debonding failure of the adhesive layer between the SMA patch and steel plate.

Except for the G8 adopted as the reference point, the other strain gauges arranged at the surface of the cracked steel plate were adopted to assist in observing the fatigue crack propagation. Taking the G6 plotted in Figure 11c, for example, it could be observed that the strain range gradually increased as the fatigue test went on. The monitored strain increased sharply once the fatigue cracks propagated close to the strain gauge. For the SP-2, the G6 was debonded rather than broken when the fatigue crack propagated below the strain gauge, and it could be seen from Figure 11c that the G6 kept working for several hours after the sharp increase in strain; however, the monitored strain at this time no longer represented the real strain of the steel plate, and it could be supposed to be broken.

### 5.2. Crack Propagation

The fatigue crack propagation life (i.e., fatigue life), as well as the crack propagation curve, were presented in this section. Regarding the fatigue life, the values corresponding to the crack propagating to 15 mm in the pre-test and corresponding to the crack propagation in the subsequent fatigue test were compared, as shown in Figure 12. It could be observed that there are few differences among the fatigue lives of SP-1, SP-2, and SP-3 in the pre-test. Compared to the fatigue life of SP-1, the one of SP-2 is 11.8% longer while the one of SP-3 is 0.7% shorter. After strengthening with the SMA patches, a considerable retarding effect on the crack propagation was obtained. The fatigue life of SP-1 in the subsequent fatigue test is 110 × 10^4^ cycles, while the one of SP-2 and SP-3 has increased by 188.9% and 168.3% (see Figure 12), respectively. It indicates that bonding the SMA patches is feasible to strengthen the crack steel plate subjected to fatigue bending loading.

Since it is difficult to measure the crack length when the crack propagates below the SMA patch, the crack length is not recorded until the crack propagates beyond the edge of the SMA patch. The crack propagation curves of SP-1, SP-2, and SP-3 were plotted in Figure 13. It is observed that the propagating rate of the fatigue crack increases with the increasing loading cycle and crack length. However, there are no significant differences among the crack propagation curves corresponding to the same side of the specimen. It indicates that the crack propagation life is less affected by the bonded SMA patch once the crack propagates out of the cover range. Therefore, integrating the data plotted in Figure 12, it could be concluded that the fatigue crack could be retarded only if it propagates within the cover range of the bonded SMA patch, whether there is debonding failure between the SMA patch and cracked steel plate or not.

### 5.3. Fracture Surface Analysis

The fatigue fracture surfaces of SP-1, SP-2, and SP-3 were obtained after fatigue tests. Figure 14 shows the fracture surface of SP-1 and SP-2 for comparison. As can be seen in Figure 14, the original place where the crack was initiated is observed on both the top and bottom surfaces of the cracked steel plate on the edge of the artificial cutting surface. However, the top crack propagation dominates the final fatigue failure. The benchmarks (i.e., the progression marks), generated by the variations in the crack growth rate, could be clearly seen in the fracture surface because the crack growth rate increases with the increasing crack length (see Figure 13).

Additionally, the fast fracture zone (i.e., the overload zone), generated when the crack reached the point where the remaining material was overstressed, could be clearly seen on both sides of the fracture surface. Regarding SP-1, the crack propagation surface in the latter stage of the fatigue life is macroscopically brittle, especially on the left side, as shown in Figure 14a: the crack propagation surface is rough, and the benchmarks can be clearly seen. It is supposed that the rough surface is generated by only a few stress cycles in the latter stage of the fatigue life. However, regarding the SP-2, the crack propagation surface is generally smooth, and the fast fracture zone is relatively small. This could be attributed to the reinforcement effect of the SMA patch, which decreases the magnitude of the load sustained by the cracked steel plate when the final fracture occurs.

## 6. Conclusions

In the present study, the feasibility of attaching the SMA patches to the reinforcement of the mode Ⅰ, mode Ⅱ, and mode Ⅲ cracks was investigated. Finite element analysis-based parametric analysis was conducted to investigate the effect of the bonding area, the thickness of the SMA patch, and the strengthening angle on the reinforcement effect. Fatigue tests were also conducted to investigate the reinforcement effect by bonding the SMA patch. The following conclusions can be drawn.

(1)Bonding the SMA patch is applicable to the mode Ⅰ crack while inapplicable to both mode Ⅱ and mode Ⅲ cracks. Compared to the strengthening method by bonding the steel patch, bonding the SMA patch could further decrease the SIF at the crack tip because of the prestress force (i.e., the recovery force).(2)The SIF has decreased 144.2 MPa·mm^1/2^ in the presence of every 1 mm thickening of the SMA patch. It is recommended that the SMA patch be arranged with the direction of the recovery force perpendicular to the possible crack propagation direction. Comparatively, the reinforcement effect is less affected by the bonding area of the SMA patch.(3)Experimental results demonstrate the retarding effect on the crack growth by bonding the SMA patches. Compared with the bare steel plate, the fatigue life of SMA patch-strengthened specimens has increased by 188.9% and 168.3%, respectively. However, the retarding effect could be neglected when the fatigue crack propagates out the cover range of the SMA patch.

## Figures and Tables

**Figure 1 materials-16-07259-f001:**
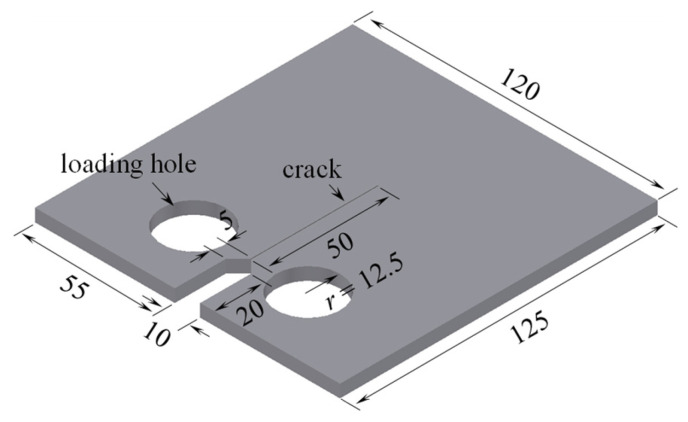
Schematic view of the CT model (unit: mm).

**Figure 2 materials-16-07259-f002:**
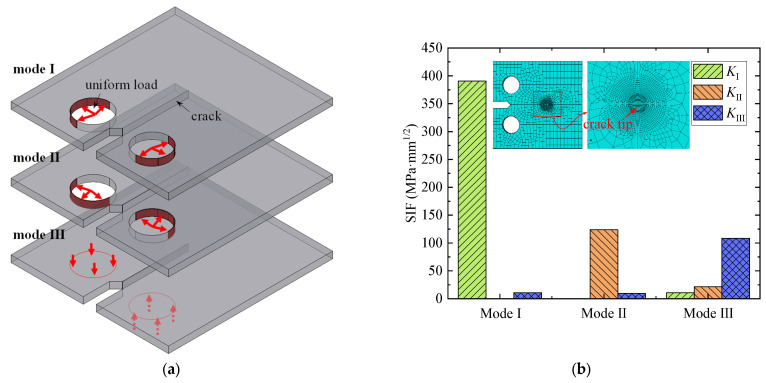
CT model and numerical results of the SIFs corresponding to the modes Ⅰ, Ⅱ, and Ⅲ cracks. (**a**) CT model simulating the modes Ⅰ, Ⅱ, and Ⅲ crack. (**b**) SIFs corresponding to the modes Ⅰ, Ⅱ, and Ⅲ cracks.

**Figure 3 materials-16-07259-f003:**
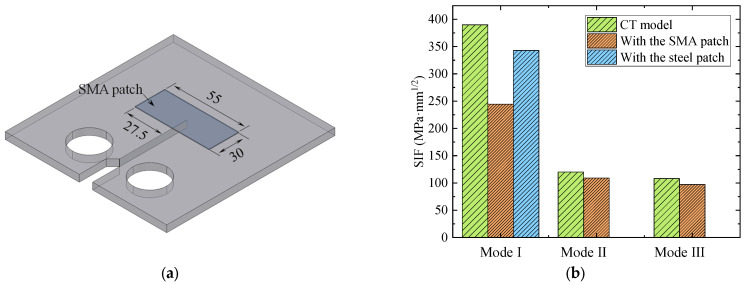
Comparison of the values of *K*_Ⅰ_ corresponding to the CT model, CT model strengthened by the Fe-SMA patch, and CT model strengthened by the steel patch. (**a**) CT model strengthened by the Fe-SMA patch. (**b**) Comparison of the values of *K*_Ⅰ_.

**Figure 4 materials-16-07259-f004:**
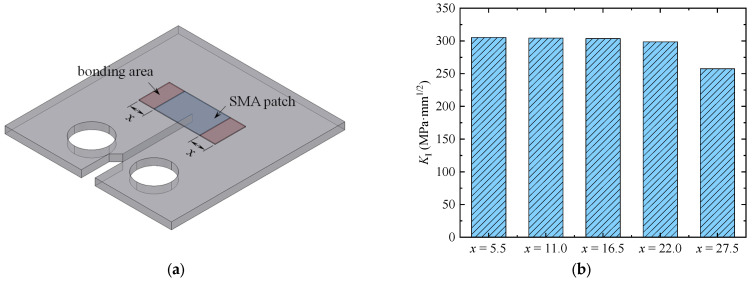
Variation in the SIFs in the presence of different bonding areas. (**a**) Illustration for the bonding area. (**b**) Variation in the SIFs.

**Figure 5 materials-16-07259-f005:**
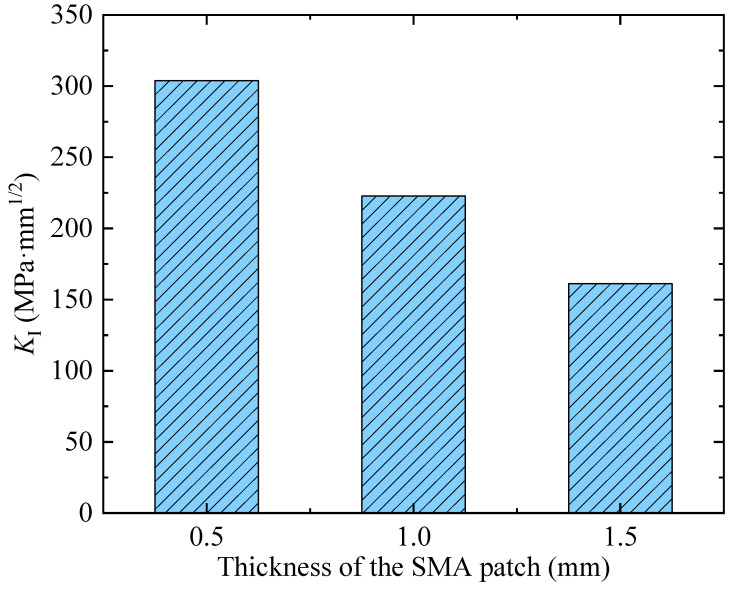
Variation in the SIFs in the presence of different thicknesses of the SMA patch.

**Figure 6 materials-16-07259-f006:**
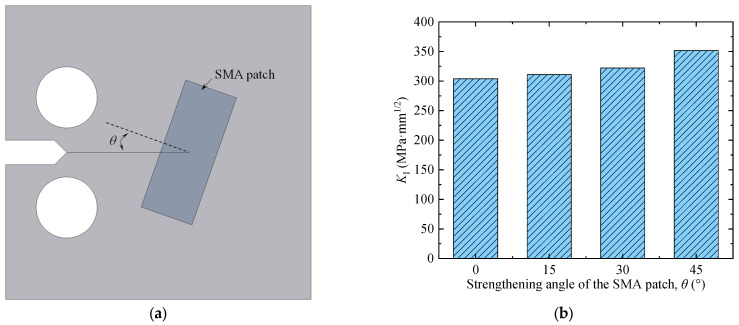
Variation in the SIFs in the presence of different strengthening angles. (**a**) Illustration for the strengthening angle. (**b**) Variation in the SIFs.

**Figure 7 materials-16-07259-f007:**
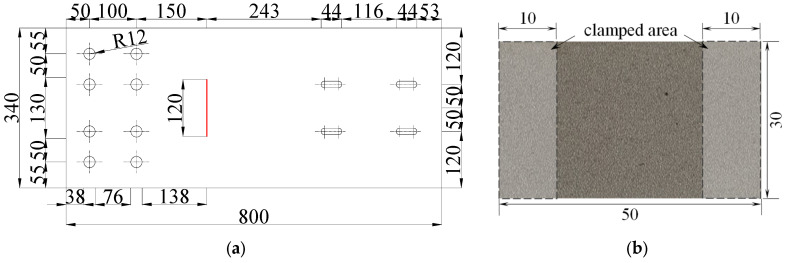
Geometric sizes of the specimen and Fe-SMA patch (unit: mm). (**a**) Tested specimen. (**b**) Fe-SMA patch.

**Figure 8 materials-16-07259-f008:**
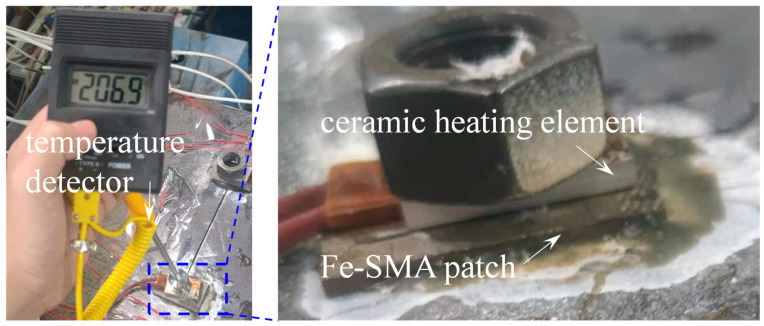
Thermal activation of the prestrained Fe-SMA patch.

**Figure 9 materials-16-07259-f009:**
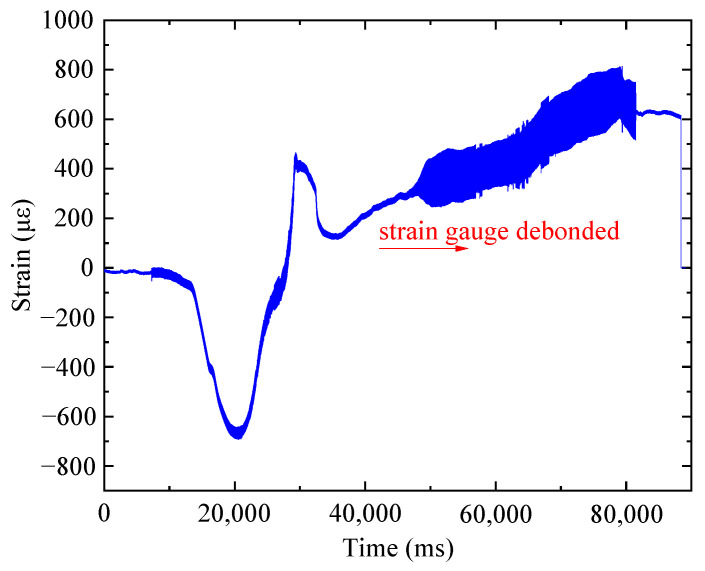
Monitored strain-time curve of the Fe-SMA patch during thermal activation.

**Figure 10 materials-16-07259-f010:**
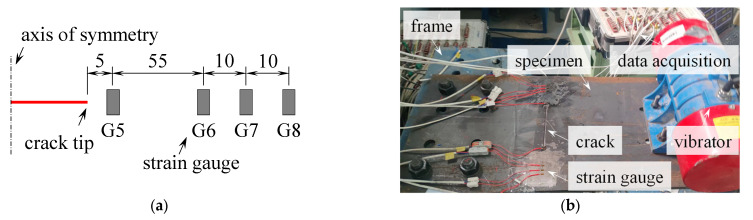
Illustration for the fatigue test. (**a**) Schematic drawing of the arrangement of the strain gauges (unit: mm). (**b**) Fatigue test setup.

**Figure 11 materials-16-07259-f011:**
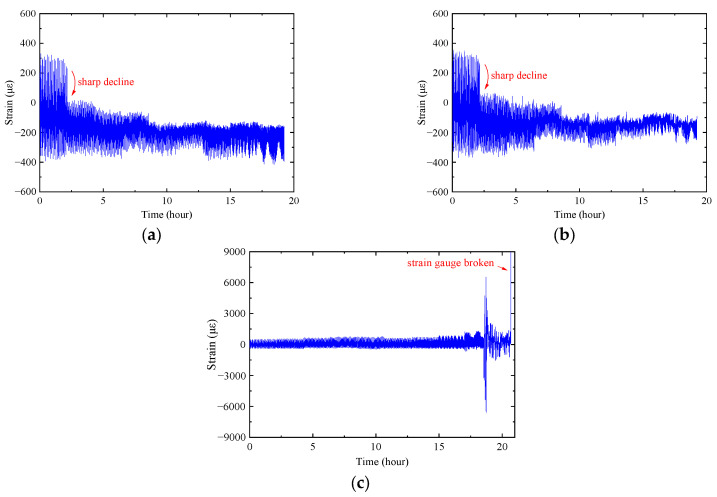
Strain-time curves of G4, G5, and G6 of SP-2 during the fatigue test. (**a**) G4. (**b**) G5. (**c**) G6.

**Figure 12 materials-16-07259-f012:**
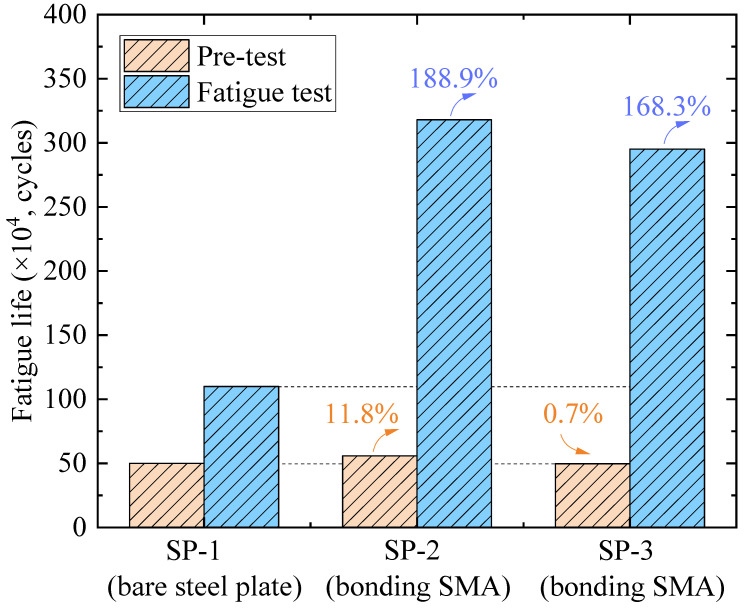
Fatigue life comparison.

**Figure 13 materials-16-07259-f013:**
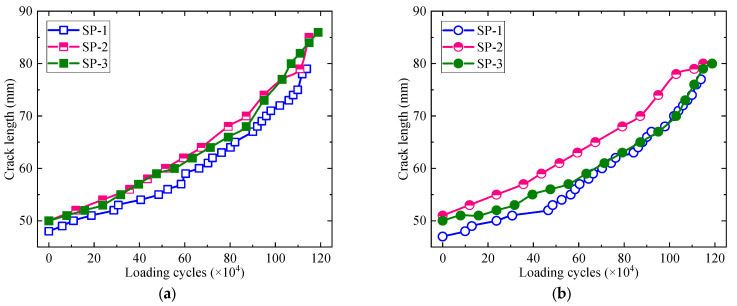
Crack propagation curves. (**a**) On the left side of the specimen. (**b**) On the right side of the specimen.

**Figure 14 materials-16-07259-f014:**
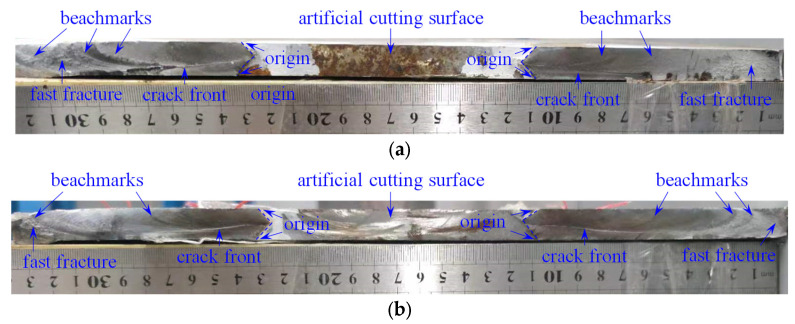
Fatigue fracture surfaces. (**a**) SP-1. (**b**) SP-2.

**Table 1 materials-16-07259-t001:** Mechanical properties and chemical composition of the Q345qD material.

Elastic Modulus (MPa)	Yield Strength (MPa)	Ultimate Tensile Strength (MPa)	Chemical Composition (%)				
2.1 × 105	345	490	C	Si	Mn	P	S
			0.14	0.31	1.46	0.016	0.006

**Table 2 materials-16-07259-t002:** Mechanical properties and chemical composition of the Fe-SMA material.

Elastic Modulus (MPa)	Yield Strength (MPa)	Ultimate Tensile Strength (MPa)	Chemical Composition (%)						
1.73 × 105	546	1015	Cr	C	V	Si	Mn	Ni	S
			8	≤0.4	≤4	5	15	5	0.006

## Data Availability

The data that support the findings of this study are available from the corresponding author upon reasonable request.

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
