# Peer review of "Strengthening Cracked Steel Plates with Shape Memory Alloy Patches: Numerical and Experimental Investigations"

_materials, 2023, doi:10.3390/ma16237259_

Round 1

Reviewer 1 Report

Comments and Suggestions for Authors

Dear Authors,

Thank you for your interesting and well-written manuscript. Please, address the comments and suggestions in the attached Word document. Best regards

Comments on the Quality of English Language

I think the technical language is clear enough. I did notice several grammar errors, so I suggest proofreading the text by a language expert (for example: “numerical and experimental analysis were” – adjust singular/plural; “the crack been detected” – has been; not “be rectangle”, but “be a rectangle”; “crack propagated closet”, and so on). I would also suggest rechecking for double (triple?) spacing between words, especially when crack-opening models are mentioned. Also: remove blanks before commas, use the proper minus sign instead of the dash, etc.

Reviewer 2 Report

Comments and Suggestions for Authors

In the paper, both numerical and experimental analyses were conducted to investigate the retarding effect of bonding the shape memory alloy (SMA) patches on crack propagation in steel plates. The monitored stress variation, crack propagation behavior, and the analysis of fatigue fracture surfaces were carried out. Compared to the bare steel plate, the fatigue life of the SMA patch-strengthened steel plates was increased by 188.9% and 168.3%, indicating the reliable reinforcement effect.

There are already some papers on the SMA patches used for improving the fatigue life of structures. In the reference list of the paper, only some papers are given on this subject. But, some very related papers are not given. The important thing, there should be enough information about how this study is presenting new information on the application and effects of SMA patches. 

The experimental setup has many similarities with the studies already published in this subject. Therefore, it is very important to explain the originality of the current paper in detail.

In the conclusion, it is written that "The thicker the SMA patch is, the better the reinforcement effect is." However, this is very general statement and normally there should be a range for the thickness of the patch.

Comments on the Quality of English Language

There are some writing errors. The English writing quality should be improved.

Reviewer 3 Report

Comments and Suggestions for Authors

The manuscript should include nomenclature - a complete list of symbols, abbreviations and markings - please add nomenclature.

The abstract at paper should be more condensed - it is a bit too long. Please shorten the abstract - it should contain only the most important information about the manuscript, without providing results and conclusions. The abstract should encourage people to read the paper.

The paper contains a good review of the literature. Congratulations to the authors in this regard.

I have no comments on the experimental part of the manuscript - it is very well described.

In terms of numerical calculations, the model description itself requires improvement. The authors provide the type of finite elements and their size - are 5 mm and 2 mm too large? In fracture mechanics and material fatigue, we deal with stress concentration. You need to focus on a mesh of finite elements in smaller sizes - this has been shown in many scientific papers. At this point, I suggest that the authors describe the numerical model very carefully.

In your paper, please show a figure of the numerical model with the assumed boundary conditions and the applied load. Please indicate whether the entire specimen was modeled, or half, or maybe a quarter or an eighth - I assume the entire specimen. Please provide more details about numerical modeling, please provide the number of numerical integration points. Please let me know how many nodes and finite elements there were in the model. How was the convergence of the numerical model tested? Please show in the figure the division into finite elements in the model for the entire specimen, as well as - especially in the vicinity of the crack. What were the finite elements there? What was the type of crack tip – how was it modeled?

Did the authors test a mesh with 8 finite elements or did they test FEA models for meshes with 20 and 27 node finite elements? What was the basis for selecting the FE model - what were the convergence criteria? Were the results compared with calculations carried out for the assumption of a plane stress state - were there any differences or not - maybe there is no need to conduct a three-dimensional analysis.

The work is interesting and can be published, but please follow my comments. I suggest minor/major revision.

Comments on the Quality of English Language

Minor editing of English language required.

Round 2

Reviewer 2 Report

Comments and Suggestions for Authors

Authors added explanations on the novelty of the paper.

Reviewer 3 Report

Comments and Suggestions for Authors

The authors took into account all my recommendations. I suggest accepting the manuscript in its current form.

Comments on the Quality of English Language

Minor editing of English language required.